# Deciphering the Functional Status of Breast Cancers through the Analysis of Their Extracellular Vesicles

**DOI:** 10.3390/ijms241613022

**Published:** 2023-08-21

**Authors:** Alexis Germán Murillo Carrasco, Andreia Hanada Otake, Janaina Macedo-da-Silva, Veronica Feijoli Santiago, Giuseppe Palmisano, Luciana Nogueira de Sousa Andrade, Roger Chammas

**Affiliations:** 1Center for Translational Research in Oncology (LIM24), Instituto do Cancer do Estado de Sao Paulo (ICESP), Hospital das Clinicas da Faculdade de Medicina da Universidade de Sao Paulo (HCFMUSP), São Paulo 01246-000, Brazil; agmurilloc@usp.br (A.G.M.C.); andreia.otake@hc.fm.usp.br (A.H.O.); lnsa@usp.br (L.N.d.S.A.); 2Comprehensive Center for Precision Oncology, Universidade de São Paulo, São Paulo 01246-000, Brazil; 3Departamento de Parasitologia, Instituto de Ciências Biomédicas, Universidade de São Paulo, São Paulo 05508-000, Brazil; janaina.macedo.silva@usp.br (J.M.-d.-S.); veronicafeijoli@gmail.com (V.F.S.); palmisano.gp@usp.br (G.P.); 4School of Natural Sciences, Macquarie University, Macquarie Park, NSW 2109, Australia

**Keywords:** extracellular vesicles, breast cancer, omics, theranostics, nanomedicine

## Abstract

Breast cancer (BC) accounts for the highest incidence of tumor-related mortality among women worldwide, justifying the growing search for molecular tools for the early diagnosis and follow-up of BC patients under treatment. Circulating extracellular vesicles (EVs) are membranous nanocompartments produced by all human cells, including tumor cells. Since minimally invasive methods collect EVs, which represent reservoirs of signals for cell communication, these particles have attracted the interest of many researchers aiming to improve BC screening and treatment. Here, we analyzed the cargoes of BC-derived EVs, both proteins and nucleic acids, which yielded a comprehensive list of potential markers divided into four distinct categories, namely, (i) modulation of aggressiveness and growth; (ii) preparation of the pre-metastatic niche; (iii) epithelial-to-mesenchymal transition; and (iv) drug resistance phenotype, further classified according to their specificity and sensitivity as vesicular BC biomarkers. We discuss the therapeutic potential of and barriers to the clinical implementation of EV-based tests, including the heterogeneity of EVs and the available technologies for analyzing their content, to present a consistent, reproducible, and affordable set of markers for further evaluation.

## 1. Introduction

Breast cancer (BC) is currently the most commonly diagnosed type of cancer in the world. In 2020, this disease accounted for 25.5% of new cases among women, and it was responsible for 15.5% of the cases of death via cancer among women [1]. Due to these rates, it is important to continue searching for screening and therapeutic options. Cancer is a disease produced by specific tissue cells with uncontrolled growth. Tumor cells use cell-to-cell communication to spread through diverse strategies, invading lymph nodes or distant organs, evading growth suppressors, or inducing tumor-supporting angiogenesis [2].

Extracellular vesicles (EVs) represent one form of cell-to-cell communication through a range of different types of membranous nano- and microparticles (ranging from 30 nm to 5 µm in size), each with a specific biogenesis, cargo, and function [3]. EVs are involved in several biological processes such as cell signaling, cell proliferation, immune system modulation [4,5,6], and the production of amphisomes as a consequence of their fusion with autophagosomes [7]. EV proteins are markers for early diagnosis and are prognostic in several types of solid tumors. In lung cancer, EV leucine-rich alpha-2-glycoprotein 1 was reported to be up-regulated in urinary and lung tissue [8]. Moreover, ovarian cancer EVs are enriched in integrin, PI3 kinase, p53, Ras, and other proteins related to cancer development [9,10]. In gastrointestinal system cancers, the amount of exosome is increased in patients with colorectal cancer, and it is correlated with carcinoembryogenic antigen [11]. EVs also participate in cancer progression processes such as tumor invasion, growth, and metastasis [12,13]. For example, several studies have shown that EV protein content is associated with breast cancer cells, demonstrating its modulation during oncogenesis and tumor progression. EVs support cancer progression, signaling recipient cells’ motility and growth [14]. In breast cancer, increased EV biogenesis has been related to the activation of protease-activated receptor 2 (PAR2), which is induced by different factors, including coagulation factor-FVIIa and trypsin [15]. PAR2 is cleaved by trypsin, which triggers PIK3-dependent AKT phosphorylation. AKT’s phosphorylation induces Rab5a activation, resulting in actin polymerization, which contributes to microvesicle budding, cell migration, and invasion [16]. Moreover, three different signaling cascades contribute to EV production via PAR2: (1) actin polymerization via the sequential activation of P38, MK2, and HSP27; (2) ERK1/2 activation, which stimulates MLC2 via MLCK; and (3) MLC2 activation via ROCK-II independently of ERK1/2 activation. The activation of MLC2 and HSP27 is essential to actomyosin rearrangement and EV production [15].

The versatility attributed to EVs is due to their heterogeneity. EVs include at least three main groups of membranous particles: exosomes, ectosomes, and apoptotic bodies [3]. Exosomes (30–150 nm in size) are initially packaged by a cell membrane bud called a multivesicular body (MVB), whereas ectosomes (200 nm–5 µm in size) are produced by protrusions of the cell membrane shed by the cell. On the other hand, apoptotic bodies are formed via cell fragmentation during programmed cell death [3]. Therefore, these particles are classified according to their sizes and the expression of a signature of surface proteins, a consequence of their different biogenesis processes. Nevertheless, the classification of these EV subtypes neglects a fourth group of EVs with no expression of the minimal set of markers described for any of the three first groups. This group includes particles whose functions and surface proteins are still being discovered, for example, migrasomes [17]. Extracellular vesicle biology is a highly dynamic and relatively recent research field. The International Society of Extracellular Vesicles (ISEV) seeks to harmonize the studies in the field, and it continuously updates the markers, mechanisms, and technologies for classifying these EVs in documents published as Minimal Information for Studies of EVs (MISEV) [18]. As researchers are still adopting the newest edition of MISEV, which includes recommendations for characterizing EV subtypes, for this review, we did not focus on specific EV subgroups, except when the authors have defined their particles as belonging to one of these subgroups.

Furthermore, EVs have arisen as a growing field of research due to their main characteristics, namely, (i) a wide of sizes from nano- to microscale (30 nm–5 µm) enabling their classification into groups (small or large EVs) [18,19,20]; (ii) a great diversity of proteins present on the surface of their lipid bilayer membranes, generally related to the molecular signature of the cell of origin [20,21]; and (iii) rhe mixed composition of their lumen or cargo, consisting of messenger RNA (mRNA) [22], microRNA (miRNA) [23], long non-coding RNA (lncRNA), circular RNA (circRNA) [24], double-stranded DNA (dsDNA) [25], mitochondrial DNA (mtDNA) [26], and proteins [27].

These characteristics allow EVs to be present in human fluids such as blood, urine, or saliva as a relevant form of cell-free circulating material helping to provide special protection for their luminal cargo, increasing the latter’s stability and half-life [28]. Furthermore, there is currently a need to understand more about the biology behind these vesicles and their potential applications, stimulating the creation of new EV-related databases [29,30,31]. These databases aim to organize the information on EV profiling being continuously produced worldwide despite the technical challenges surrounding their isolation and characterization [32].

Since EVs are collected through minimally invasive methods and carry different types of information, these particles represent a suitable source of information that may lead to the improvement of breast cancer screening and treatment. However, we still face challenges related to (i) the requirement for the proper identification and quantification of EVs; (ii) the identification of breast-cancer-related markers in EVs; and (iii) strategies for taking advantage of these breast-cancer-related EVs to create new therapeutic options, among others (Figure 1).

Transcriptomic approaches are usually performed on EVs secreted by cell lines in vitro and also in body fluid samples, which could be used to select targets for regulation in tumors. Meanwhile, since exploratory proteomic approaches are conventionally performed using cell lines, we can use this approach to characterize EVs from a single cell type from a classic perspective and propose tools for recognizing tumor-derived EVs. Therefore, this review aims to support the development of strategies for overcoming these challenges by compiling proteomic and transcriptomic information previously described in relation to breast cancer-derived EVs.

## 2. Transcriptomic and Genomics of BC-Related EVs

In addition to proteins, nucleic acids are another important component of EVs. The nucleic acid profile of Evs is primarily characterized by RNA types such as mRNA [24,33,34,35], miRNA [36,37,38,39], circRNA [24,40], rRNA [41,42], siRNA [41], and lncRNA [24,43]. However, mtDNA [33,44] and viral DNA [45] were also described in EVs from breast cancer samples. Herein, we explored the transcriptomic and genomic profiles of BC-related EVs to search for common markers and specific signatures associated with breast tumor features.

### 2.1. miRNA Profile in BC-Related EVs

Extracellular RNA (exRNA) is found in body fluids and can be protected from RNases through binding to circulating proteins like Ago2 and being sorted into extracellular vesicles (EVs) [23,46]. Since it is known that cancer cells secrete heterogeneous populations of EVs and that these EVs indeed contain diverse species of functional RNAs [47], the concept of accessing tumor RNA using circulating EVs is gaining great attention in the oncological field.

Among all vesicular nucleic acids, miRNAs represent the largest (and most analyzed) group. So far, at least 128 vesicular miRNAs are being studied for breast cancer (Appendix A [36,38,39,48,49,50,51,52,53,54,55,56,57,58,59,60,61,62,63,64,65,66,67,68,69,70,71,72,73,74,75,76,77,78,79,80,81,82,83,84,85,86,87,88,89,90,91,92,93,94,95,96,97,98,99,100,101,102,103,104,105,106,107,108,109,110,111,112,113,114,115,116,117,118,119,120,121,122,123]). These studies comprise analyses conducted to characterize breast tumors as well as understand the tumorigenic process and propose candidate vesicular biomarkers for diagnosis and prognosis.

Despite the high diversity described for this type of RNA, EVs from breast cancer culture supernatants (Appendix A, Figure 2A) and liquid biopsies (Appendix A, Figure 2B) have been shown to have three miRNAs in common that can be potential markers of this tumor: hsa-miR-21, hsa-miR-122, and hsa-miR-1246.

Interestingly, the miRNAs hsa-miR-21 and hsa-miR-122 were found in EVs from cell lines belonging to all the main subtypes of breast cancer following the classification provided by Dai et al. (2017) [124]. Nevertheless, it is essential to note that this review includes studies presenting both exploratory (RNA-seq, microarray, barcoding hybridization, etc.) and targeted (qRT-PCR) experiments analyzing samples from different countries/ancestral groups. Targeted analyses have broadly described oncomiRs, e.g., hsa-miR-21 [125,126] (Figure 3). Subsequently, this miRNA was extensively tested in extracellular vesicles, with the results confirming their role as an inductor of tumor processes [36,37,38,65,72,110] despite the existence of a minority of studies with divergent results [90]. In addition, Arisan et al., 2021 demonstrated that vesicular hsa-miR-21 facilitates the expansion of breast cancer cells favoring the epithelial–mesenchymal transition (EMT) by targeting Wnt-11 [66]. Moreover, this miRNA can support bone metastasis via the inhibition of PDCD4 [36,110] and cancer-related thrombosis by blocking IL6R in endothelial progenitor cells [65]. Regarding their involvement in the tumor microenvironment, Donnarumma et al. (2017) suggested that hsa-miR-21 could be transferred from cancer-associated fibroblasts (CAFs) to breast cancer cells via exosomes [127]. The horizontal transfer from this miRNA creates a self-regulating cycle that drives the presence of hsa-miR-21 in tumor cells.

On the other hand, other microRNAs like hsa-let-7f, hsa-miR-221, and hsa-miR-770 have consistently been found in EVs from Triple-Negative BC (TNBC) cells (TNA and TNB groups, Figure 2A). Interestingly, Wei et al. (2014) showed that vesicular hsa-miR-221 induced tamoxifen resistance in MCF7 cells (Luminal A subtype) by repressing the expression of p27 and ERα [76]. However, if an estrogen-receptor-positive BC cell receives hsa-miR-221 from an estrogen-receptor-negative cell through EVs, it induces the repression of the receptor via the MAPK pathway [128] (Figure 3). In TNBC, this vesicular miRNA will induce an EMT by targeting PTEN while also stimulating the expression of mesenchymal markers such as Snail, Slug, N-cadherin, and vimentin [75]. These findings support the hypothesis that specific components of vesicular cargo can modulate the aggressiveness of cell lines.

Nevertheless, one should keep in mind that these findings raise some concerns about the similarity of cargoes between EVs produced by pure cell lines and those captured via liquid biopsies that are secreted by multiple and heterogeneous types of cells, including breast tumors and non-malignant cells such as platelets. Among all the vesicular miRNAs observed in body fluids (Figure 2B), there is still a notable difference between miRNAs found in EVs from plasma and serum. Although these sources can be separated from blood, they have been reported to include different EV-related compositions [129,130] and, according to this review, varied cargoes as well. Intriguingly, hsa-miR-21 and hsa-miR-188 were consistently found in EVs from the plasma and serum of BC patients (Figure 2B). In addition, the study conducted by Inubushi et al. (2020) described the presence of hsa-miR-21 in the tears of BC patients [118]. Since cancer research is moving toward analyzing liquid biopsy sources, these efforts are intended to improve the screening and selection of candidates who will undergo solid biopsies (which are highly invasive). More recently, the scientific community has been interested in classifying studies using systemic liquid biopsy sources (blood, plasma, or serum) and local liquid biopsies (tears or urine), with the latter group guaranteeing completely noninvasive tests. Therefore, further research in this field of study is warranted in order to assess its feasibility.

A frequently encountered difficulty in characterizing EVs from body fluids concerns the specific tumor subtype of each patient cohort (Figure 1). In our review, we identified hsa-miR-373, hsa-miR-296, hsa-miR-32, and hsa-miR-548d that were exclusively expressed in EVs from the serum of TNBC patients. In addition, hsa-miR-373 was found in the EVs of luminal BC patients [113]. A few years later, hsa-miR-27a/b, hsa-miR-335, hsa-miR-365, hsa-miR-376c, hsa-miR-382, hsa-miR-422a, hsa-miR-433, and hsa-miR-628 were found to be differentially expressed in plasma small vesicles (classified by the authors as exosomes) from either TNBC or HER2+ patients compared to healthy individuals. Moreover, the authors found that a specific set of microRNAs composed of hsa-miR-16, hsa-miR-328, and hsa-miR-660 was associated with lymph node status among HER2+ patients, indicating that vesicular microRNA can also be associated with clinicopathological characteristics [131]. Interestingly, based on the increased abundance of hsa-miR-150-5p, hsa-miR-576-3p, and hsa-miR-4665-5p in exosomes from the plasma of BC patients, Wu et al. (2020) were able to distinguish patients with recurrence from patients without disease recurrence [132]. Finally, in this review, we identified the following vesicular miRNAs present in patients from all BC subtypes: hsa-miR-122, hsa-miR-16, hsa-miR-93, hsa-miR-423, and hsa-miR-101. Tissue-specific alterations are associated with different expression profiles such as HER-2 overexpression [121], TNBC [91,133], or the stem-like profile among basal-like ductal carcinoma in situ (DCIS) tumors [134], demonstrating the possibility of modifying levels of vesicular miRNAs. Once these findings have been consistently reported by studies performed on different BC cohorts, we believe these miRNAs deserve further attention as potential BC biomarkers in EVs.

In addition, short non-coding RNAs, including miRNA and siRNA, are potential therapeutic options. In this case, it is possible to transfect nucleic acids in extracellular vesicles to induce transcriptional changes in recipient cells. For example, the suppressor miRNA hsa-let-7a was used to design anti-tumor extracellular vesicles for targeting EGFR [68] and c-Myc [79] to repress tumor growth. Subsequently, Bose et al. (2018) suggested editing tumor-derived EVs from SKBR3 BC cells so that they included Cy5-anti-miR-21 in their cargo. This technique allowed for the inhibition of the oncogenic hsa-miR-21 and the tracking of the biodistribution of these tumor-derived EVs [135]. Afterward, the same group developed an engineered version of Evs coated on polymeric nanocarriers and loaded with anti-miR-21 and anti-miR-10b [70]. The prototype was tested in an in vivo model of TNBC that demonstrated an improved biodistribution to the tumor and the ability to exert a synergic effect with doxorubicin to control tumor growth.

Following this concept, Taghikhani et al., 2018 modified the cargo of tumor-derived extracellular vesicles so that they included higher concentrations of hsa-miR-155, hsa-miR-142, and hsa-let-7i for the further treatment of immature dendritic cells [136]. Their results showed that vesicular hsa-let-7i, enhanced by hsa-miR-155 and hsa-miR-142, acted as a cell-free vaccine for cancer treatment by inducing the maturation of dendritic cells. Using this characteristic, some studies have proposed the modification of autologous BC-derived EVs including siS100A4 [137] and ASO-1537S [138]. siS100A4 targets S100A4, a previously reported gene for the EMT in cancer [139], whereas ASO-1537S is an antisense oligonucleotide against 1537S, a non-coding mitochondrial RNA identified in oncogenic processes [138]. After these insertions, modified EVs prevented metastasis in an in vivo model for BC, providing a non-immunogenic method of transferring information and controlling cancer.

On the other hand, research on suppressor miRNA that is lowly expressed in tumor-derived EVs also opens a new window for therapy development. According to Zhou et al. (2021), BC cells treated with EVs carrying hsa-miR-424-5p, a suppressor miRNA, inhibited their growth though the miRNA-driven repression of PD-L1 [140].

Amalgamating these precedents, researchers have proposed new tools for cell editing using EVs (Figure 1). The main idea is to select (or develop) specific subpopulations of EVs that we can use to target cancer cells or change their environments. Nevertheless, although the use of vesicular microRNAs for diagnosis and prognosis seems promising, no panels of EV microRNAs are available for BC diagnosis in a clinical setting due to a lack of standardization and reproducibility concerning EV isolation and microRNA identification, as discussed in the final section (Figure 1).

### 2.2. Other Regulatory Non-Coding RNAs in BC-Related EVs

As described above, vesicular miRNAs can inhibit mRNA expression in their target cell. Then, oncomiRs repress tumor suppressors, and suppressor miRNAs inhibit the expression of tumor promoters. Nevertheless, exploratory techniques such as RNA-seq have supported the discovery of competing endogenous RNA (ceRNA) in BC-derived EVs [24,43]. This type of RNA can attract miRNAs by competing with their mRNA targets, thereby adding a new level of EV-based expression regulation. Herein, a ceRNA can inhibit a suppressor miRNA in order to induce the overexpression of the mRNA target, which stimulates tumor growth.

Some lncRNAs, a type of ceRNA, have been reported to be overexpressed in EVs from BC tissues, cells, and human fluids. For example, the lncRNAs SNHG16 [141,142], HOTAIR [143], H19 [144], SNHG14 [145], AFAP1-AS1 [146], AGAP2-AS1 [147,148], BCRT1 [149], UCA1 [150], GS1-600G8.5 [151], and NEAT1 [152] were found to be overexpressed in BC-derived EVs (Figure 3). Moreover, they have demonstrated their ability to induce new features in recipient cells.

SNHG14, AFAP1-AS1, AGAP2-AS1, and BCRT1 have been proven to be carried by tumor-derived EVs in order to induce trastuzumab resistance in BC cells (Figure 3). Interestingly, they achieve the same outcome in different ways. SNHG14 affects the Bcl-2 apoptosis pathway [145]; AFAP1-AS1 stimulates the overproduction of HER2, leading to treatment ineffectiveness [146]; AGAP2-AS1 is packaged in vesicles by hnRNPA2B1 [148] to link ELAVL1 in the receptor cell in order to modulate autophagy activity via the transcription of ATG10 [147]; and BCRT1 inhibits hsa-miR-1303 in order to induce the expression of PTBP3, a tumor promoter and target of hsa-miR-1303 [149]. Moreover, other studies have shown that vesicular H19, NEAT1, and UCA1 lncRNAs induce resistance to doxorubicin [144], cisplatin/paclitaxel [152], and tamoxifen [150], respectively. In all cases, it is possible to revert treatment resistance by silencing the respective lncRNA.

Tumor growth and metastatic expansion are processes that involve many molecular regulations. Accordingly, the mRNA–miRNA–lncRNA axis is analyzed to understand cancer biology. This reseaerch explains how tumor-derived EVs help to disseminate a competitive advantage among cancer cells. However, in some cases, they can “educate” neighboring epithelial and immune cells in terms of facilitating tumor expansion. For example, vesicular SNHG16 lncRNA induces metastasis by blocking hsa-miR-892 in BC cells; this action naturally represses PPAPDC1A, a metastasis promoter [142]. But the same lncRNA is able to block hsa-miR-16-5p in γδT1 lymphocytes in order to induce SMAD5 expression, which upregulates CD73 at the membrane of these tumor-infiltrating lymphocytes [141]. Vesicular BCRT1 induces M2 polarization in macrophages, thereby boosting BC progression [149]. Also, the presence of vesicular GS1-600G8.5 was associated with the destruction of the blood–brain barrier, thus facilitating brain metastases in BC patients [151].

In the same way that vesicular lncRNA from BC cells can modify the metabolism of other cell types, EVs produced in other cell types affect tumor growth. As an example, BC-modulating vesicular HISLA and SNHG3 lncRNAs are expressed in tumor-associated macrophages (TAM) and cancer-associated fibroblasts (CAF), respectively. HISLA obstructs the interaction between PHD2 and HIF-1α that leads to the accumulation of HIF-1α, which regulates glycolytic metabolism in the tumor cell [153]. SNHG3 sponges hsa-miR-330-5p in tumor cells that positively regulate PKM, thereby increasing tumor growth [154]. However, the transcriptome from BC-derived EVs also includes suppressor lncRNAs such as XIST, whose repression promotes brain metastasis in BC patients [155]. The absence of XIST induces an accumulation of hsa-miR-503, which triggers M1-M2 polarization, thereby suppressing T-cell proliferation and, in turn, facilitating tumor expansion.

For the time being, we have limited information regarding circRNAs. Some descriptive studies have described circHIPK3 [156], circHIF1A [157], hsa-circRNA-000615 [158], hsa-circRNA-0005795, and hsa-circRNA-0088088 [159] as being overexpressed in BC-derived EVs playing vital roles in BC development (Figure 3). Interestingly, circHIPK3 was associated with trastuzumab resistance [156]. Moreover, circHIF1A was identified as an oncogenic RNA because it downregulates hsa-miR-149-5p levels [157]. Subsequently, low levels of hsa-miR-149-5p induce overexpression of NFIB, an oncogenic promoter that functions via p21 inhibition.

### 2.3. Messenger RNA (mRNA) in BC-Related EVs

Besides regulating RNA, some studies have shown that mRNA is present in EVs derived from or targeting BC cells. A few years ago, Conley and co-authors identified an mRNA signature of this disease in EVs from breast cancer patients at advanced stages [34]. Under the same context, Rodríguez et al. (2015) found elevated levels of eight metastasis-related mRNAs and 27 stemness-related mRNAs in exosomes from breast cancer patients with poor prognosis, suggesting that exo-mRNA can be used as a prognostic marker for BC patients [160].

Andreeva et al., 2021 have demonstrated that *PIK3CA* mutations can be transferred between BC cell lines via their EVs [33]. Using RNA and DNA sequencing, this study described the presence of mutated versions of the PIK3CA gene in EVs from BC cell lines. BC-derived EVs have been observed overexpressing some mRNAs. According to Rodríguez et al. (2015), vesicular levels of *HTR7*, *NEUROD1*, and *HOXC6* mRNA are related to disease-free survival among BC patients (*p* < 0.05), while *NANOG*, *HTR7*, *NEUROD1*, and *HOXC6* mRNA levels are associated with overall survival (*p* < 0.05) [160]. Moreover, vesicular *ERCC1* mRNA could be indicative of tumor progression (Figure 3) via their association with metastatic BC (*p* = 0.03) [161]. Furthermore, mRNA transcripts packaged inside tumor-derived EVs can be used as a disease indicator through liquid biopsy as proposed by Hu et al. (2023) [162]. To achieve this, the authors propose isolating different subsets of EVs using mAbs targeting tumor epitopes. Subsequently, they demonstrated that the mRNA profiling results from PAM50 genes of specific subtypes of vesicles from breast cancer patients were 100% in accordance with the tumor tissue and could be used to predict BC subtypes using minimally invasive methods.

On the other hand, different cell types can transmit growth signals to BC cells via EVs. Yao et al., 2019 demonstrated that platelets from BC patients overexpress *TPM3* mRNA and transfer it via EVs to BC cells in order to promote their growth [163].

However, it should be mentioned that these studies face challenges relating to the low abundance of vesicular RNAs and the lack of housekeeping RNA species with which to make comparisons between different samples. Indeed, these important issues push scientists in the field to discuss the standardization of and guidelines for EV RNA analysis (Figure 1). Within this context, Gorji-Bahri et al., 2021 have proposed using the *YWHAZ* gene alone or in combination with *GAPDH* and *UBC* as stable values for vesicular RNA quantification [164].

### 2.4. DNA Profiles in Breast Cancer-Derived EVs

Interestingly, besides the presence of PIK3CA mRNA in BC-derived EVs, Andreeva and co-authors also found fragments of mutated DNA in these nanocompartments [33]. Accordingly, one might assume that EVs can contribute to the discarding or transmission of pathogenic variants present in one cell to another. Ruhen et al. (2021) also demonstrated that tumors export Copy Number Variations (CNVs) through EVs. Although the transference of somatic mutations has been reported before in the form of circulating tumor DNA (ctDNA), some of these mutated copies are sheltered by EVs that allow for their direct detection using highly sensitive technologies [44].

Moreover, some studies have described the presence of mtDNA in BC-derived EVs. Fragments of mtDNA in BC-derived EVs are associated with invasiveness via the activation of Toll-like receptor 9 [165], chemoresistance in TNBC cells [166], and escape from metabolic quiescence in hormone-therapy-resistant BC via estrogen-receptor-independent oxidative phosphorylation [167]. As it has been shown, this is a hitherto unexplored field that needs more research, mainly to determine the genes responsible for mtDNA transmission that can serve as candidates for gene therapy (Figure 1). Although the *PINK1* gene was identified as a driver of mtDNA-carrying EVs [165], there are likely other drivers of this feature.

In addition, non-human DNA was also reported in BC-derived EVs. This is the case of HPV-positive EVs isolated from the serum of BC patients [45]. There is an important finding that could extend liquid biopsy features for the determination of viral subtypes in EV from patients as demonstrated by De Carolis et al., 2019 [45]. Although vesicular RNA has shown better information delivery characteristics than DNA [168], EVs serve as important tools for cellular communication that evades immune control. Therefore, EVs, their molecular topography, and their cargo deserve further attention.

Overall, the nucleic acid composition of BC-derived vesicles is highly diverse. Furthermore, some of these components are related to tumor processes such as the modulation of aggressiveness, the EMT, the preparation of the metastatic niche, and treatment resistance. Although EVs cannot deliver many nucleic acids per particle [22] (Figure 1), we present an overview of the relevant nucleic-acid-based markers in Figure 3.

## 3. Proteome of EVs in Breast Cancer

### 3.1. Analysis of BC-Related EV Proteomes Using High-Throughput Technologies

In order to elucidate the role of EV-related proteins during breast cancer pathogenesis, we reviewed the literature to determine the proteomes of EVs associated with breast cancer. A total of 21 studies [14,169,170,171,172,173,174,175,176,177,178,179,180,181,182,183,184,185,186,187,188] that analyzed EVs from human breast cancer cell lines and biofluids using high-throughput and exploratory technologies were selected. A total of 8312 proteins were reported. The proteins were divided into core level 1 (protein identification in 50–100% of studies), core level 2 (protein identification in 40–50% of the studies), and core level 3 (protein identification in less than 40% of the studies), and they are reported in Appendix A [14,169,170,171,172,173,174,175,176,177,178,179,180,181,182,183,184,185,186,187,188]. Six and twenty-six proteins were assigned to cores 1 and 2, respectively. Although these data indicate high variability in the qualitative proteome content of EVs associated with breast cancer (Figure 1), we selected a few studies that evaluated methodological strategies for isolating EVs from breast cancer cells and biofluids and characterized their protein content. As many proteomic results were produced using BC cell lines, we used the classification system developed by Dai et al. (2017) to standardize the nomenclature and disease subtypes potentially represented by these cell lines [124].

Harris et al., 2015 investigated the subcellular localization of proteins found in the EVs of three BC cell lines with different types of metastatic profiles, namely, MDA-MB-231 (Triple-negative B subtype), MCF-7 overexpressing Rab27b, and MCF-7 (Luminal A subtype), and found several cellular localizations (the majority were from cytoplasm/cytoskeleton and integral/peripheral membrane proteins) and a minor proportion of EV proteins from the Golgi apparatus, ER, or mitochondria [14].

Regarding the molecular functions of the EV proteins, the majority displayed a protein-binding function and hydrolase activity. Comparing the EV protein expression between the three cell lines, 85 proteins were differentially expressed, with the non-invasive breast cancer cell line (MCF-7, Luminal A subtype) containing several up-regulated proteins with a tumor suppression function. Among the up-regulated proteins in MCF-7 compared to MDA-MB-231 were Tetraspanin (CD63, CD81, CD9, and Tetraspanin-14), Adhesion proteins (Neural cell adhesion molecule 2, Integrin alpha-V, Integrin beta-5, Epithelial cell adhesion molecule, Alpha-Parvin, Claudin-3, Cadherin-1, CD99, CD276, and Clusterin), stress response proteins (Heat shock protein beta-1), Small GTPase superfamily proteins (Rho-related GTP-binding protein RhoB, GTPase NRas, Ras-related protein Rap-2c, and RacGTPase-activating protein 1), and Endosome Trafficking/Transport proteins (Multivesicular body subunit 12B, Vacuolar protein sorting-associated protein 28 homolog, Vacuolar protein sorting-associated protein 37B, and Multivesicular body subunit 12A) [14]. The up-regulated proteins in MDA-MB-231 and MCF-7 overexpressing Rab27b compared to wild-type MCF-7 were adhesion/motility/cytoskeleton proteins (Vimentin, Galectin-3-binding protein, Annexin A1, Plectin, and Filamin-B), cell-surface receptor proteins (Ephrin type-A receptor 2), and stress response proteins (Protein NDRG1, Stress-70 protein, mitochondrial, and heat shock protein HSP 90-beta). Many of these proteins that were dysregulated in both cell lines are associated with tumorigenesis and metastasis [14] (Figure 3). Another protein of the Rab family, Rab5a, was associated with high migration and invasion rates in MDA-MB-231 cells along with the promotion of high rates of vesicle shedding. Interestingly, this process could be externally regulated by activating protease-activated receptor 2 (PAR2), using trypsin or coagulation factor-FVIIa to promote the accumulation of Rab5a [15,16].

Kruger et al., 2014 performed a molecular characterization of EVs from MCF-7 and MDA-MB-231 BC cells [188]. Using sucrose gradient ultracentrifugation and LC-MS/MS, the authors identified 59 and 88 proteins in MCF-7 and MDA-MB-231 EVs, respectively. The identified proteins were grouped based on molecular functions, such as catalytic activity, protein transport, adhesion, and extracellular matrix activity. Both EVs isolated from MCF-7 and MDA-MB-231 supernatants presented differences in their protein content. The MCF-7-derived EVs showed a greater abundance of biomolecule-binding and protein transport activity, while the MDA-MB-231-derived EVs contained proteins with catalytic activity. Moreover, 24% of the EV proteins of MDA-MB-231 were extracellular matrix proteins. These findings can be correlated with the higher metastatic potential of MDA-MB-231 (Triple-negative B subtype) compared to MCF-7 cells (Luminal A subtype) [188].

This study reported several proteins identified in both MDA-MB-231 and MCF-7 EVs, such as proteins from the Annexin family (Figure 3), Histone H4, and Calmodulin. Furthermore, the common EV proteins participate in cellular growth, signaling pathways, epigenomic alterations, DNA and histone methyltransferase, and Akt pathway regulation [189]. Moreover, these processes can be related to the malignancy of cancer cells and the poor prognosis of breast cancer [190]. Heinemann et al., 2014 used the MDA-MB-231 breast cancer cell line to develop a simple and efficient method for purifying cancer exosomes [183]. A three-step filtration method was implemented as the first step of a normal flow prefiltration using a 0.1 µm filter to remove larger constituents such as intact cells and cell debris. In the second step, tangential flow filtration using a 500kDa membrane filter was performed in order to remove proteins, and the retentate was passed to the third step, namely, track-etch filtration, where low pressure is applied using a 0.1 µm filter to isolate exosomes and remove microvesicles. A total of 60 unique proteins were identified sequential filtration, including the exosome marker CD63. In the exosome content, membrane trafficking proteins and proteins associated with transcription regulation, signal transduction, and the epigenetic modulation of nucleic acids were identified [183]. Regarding the heterogeneity of the vesicle entities, it is important to mention that filter-based techniques usually require the use of an automatized mechanism in order to avoid user interference during vesicle collection (Figure 1). Nevertheless, these techniques seem to support a relevant volume of EV proteomic data, so their application to EVs from different cell lines is recommended.

Smyth et al., 2014 examined the exosome proteomes from MCF-7 and PC3, cell lines of mammary gland/breast, and prostate cancers [185]. The identified exosome-delivered MCF-7 proteins were Heat Shock Proteins 70, 90, and 27; CD9; Ras-related protein 13 (RAB-13 and others); Annexins 1, 2, 5, and 7; Pyruvate kinase (PKM); Alpha-enolase (ENO1); Glyceraldehyde-3-phosphate dehydrogenase (GAPDH); Glucose-6-phosphate 1-dehydrogenase (G6PD); and actin. The functions of the exosomal proteins were related to amino acid transport, protease inhibition, the cytoskeleton, GTPase, ATPase, transcription regulation, transduction, adhesion, and chaperoning. Regarding subcellular localization, most of the proteins were in the membrane and cytoplasm; however, proteins localized in the Golgi and nucleus were also identified in MCF-7 exosomes. Moreover, it was found that the exosomal lipid composition also contributes to facilitating adherence/internalization in recipient cells [185]. The current stage of analytical methods, together with the optimization of pre-analytical factors and computational strategies, allowed for a better understanding of the EVs lipid compositions [191]. A comparison of the lipid compositions between EVs and cells of origin [192] has shown that, in vitro, EVs are more enriched in cholesterol, sphingomyelin, glycosphingolipids, and phosphatidylserine compared to their origin cells, while phosphatidylcholine and phosphatidylinositol are more enriched in cells compared to exosomes. Quantitative analyses of oxysterols in exosomes released from breast cancer cells revealed that levels of 27-Hydroxycholesterol were higher in exosomes from MCF-7 cells compared to MDA-MB-231 and non-cancerous cells, showing a dependence of the levels of this exosomal lipid on the ER status of the cell of origin [193]. Another study compared the lipid profiles of EVs isolated from TNBC cell lines, namely, D3H2LN and D3H1, with high and low metastatic potential, respectively [194]. The exosomal levels of unsaturated diacylglycerols isolated from the highly metastatic cells were higher and stimulated angiogenesis through the protein kinase D signaling pathway.

The EVs isolated from the MCF10 and MDA-MB231 cell lines shared proteins such as regulators of cell death, membrane components, adhesion, and cell motility proteins. The EVs from MDA-MB-231 were enriched in proteins associated with transcriptional regulation, proteolysis, EV formation (annexin, LAMP-1), cell cycle (NUMA1), and adherence to extracellular matrices (EDIL3, collagen, vitronectin). These results showed differences in the EV protein content of invasive and non-invasive breast cancer cells, which can contribute to the metastatic process [171].

The EV proteomes of two tumorigenic breast cancer cell lines, invasive (SKBR3, Her2 subtype) and non-invasive (MCF-7), were compared to non-tumorigenic MCF-10a. In this study, the authors used a synthetic peptide (Vn96) with a high affinity for heat shock proteins (highly expressed in cancer cells) to isolate EVs. Heat shock proteins are present on the exosome surface, binding to Vn96 and facilitating EV recovery [195]. In total, 392 (SKBR3) and 301 (MCF-7) exosomal proteins were identified, and they were associated with membranes (19%) and the cell surface (12%). The functions of EV proteins from SKBR3 are related to metabolism (enolase, fatty acid synthase, phosphoglycerate kinase, fructose bisphosphatase 1, GAPDH, malate dehydrogenase, L-lactate dehydrogenase, aldehyde dehydrogenase, aldolase, triosephosphate isomerase, and glucosidase 2 subunit beta), binding (Selenium-binding protein 1, 60 kDa heat shock protein, Protein disulfide-isomerase, Lamin A/C, and Tumor protein D52), and assembly (Myosin-9, alpha-Actinin-4, Cytokeratin 16, Cytokeratin 18, Cytokeratin 8, and Cytokeratin 19). Although MCF-7 EV protein content displayed the same molecular function as SKBR3, different proteins were identified, namely, Aldolase, Pyruvate kinase, Tryptophan-tRNA ligase, Cathepsin D, Kynureninase, TER ATPase, Lactoferroxin-C, and Hexokinase-1, which are involved in metabolism, and HSP90-a, Agrin, and protein SET are related to binding. Cytokeratin 19, which was identified in both SKBR3 and MCF-7 EVs, has been suggested to be involved in more aggressive tumor proliferation, metastasis, and invasion [196] (Figure 3). Cytokeratin 8 and 18 were reported to be secreted cancer biomarkers when detected in the serum of breast cancer patients [197]. EV proteins from SKBR3 (Her2 subtype) and MCF-7 (Luminal A subtype) cells are involved in lipogenesis, glycolysis/gluconeogenesis, and tricarboxylic acid, corroborating the important role of altered metabolism in cancer. Moreover, glycolytic enzymes can protect cancer cells from stress by inhibiting apoptosis [198]. Taken together, these data show that EV cargo is associated with the progression, proliferation, metabolic control, and malignancy of breast cancer (Figure 3).

Koh et al., 2021 analyzed the modulation of the plasma EV proteome of breast cancer patients with and without cognitive impairment following anthracycline-based treatment [187]. Plasma from the early-stage breast cancer patients was collected longitudinally at three different time points (before the start of chemotherapy, 3 weeks after cycle 2 of chemotherapy, and 3 weeks of the last cycle of chemotherapy) from two groups (cognitive non-impaired and impaired). Circulating EVs were isolated through ultracentrifugation and analyzed using Tandem-Mass-Tag (TMT)-based quantitative proteomics. A total of 517 regulated proteins were identified in the cognitive impairment group, which was compared to a non-impairment group at three time points. Among these regulated proteins were EV markers, such as CD9, TSPAN14, and CD5L. Moreover, a down-regulation of galactosylceramidase from the plasma EV protein content was observed in the cognitive non-impaired group at timepoint T3 compared to T1. On the other hand, p2X purinoceptor and cofilin-1 were up-regulated at timepoint 3 compared to T1. In contrast, the EV content of the cognitive impairment group displayed a down-regulation of p2X purinoceptor, cofilin-1, nexilin, and ADAM10 at T3 compared to T1 [187]. In this study, the authors also analyzed N-glycosylation sites in the EV content from plasma samples. However, none of the regulated proteins were identified in the N-glycosylation analysis. However, CD5L, which is an EV marker, displayed two glycosylation sites in both groups (non-impaired and impaired). An altered N-glycosylation profile was identified in CD5L peptides, which presented decreased glycosylation in the non-impaired group. These findings contribute to our understanding of how different omics characteristics interact between EVs, their cells of origin, and their destination. Nevertheless, due to the heterogeneity of the tumor itself as well as the intra and interpatient heterogeneity (Figure 1), further studies would need to include more available information about the clinical statuses of these patients (age at diagnosis, molecular subtype, immunohistochemical profiles, etc.) in order to find patterns in the EV-related markers using large-scale cohort analyses.

The EVs released from cancer cells can regulate specific biological processes in recipient cells and modulate the central nervous system, thereby impairing the cognitive abilities of breast cancer patients [187]. Jordan et al., analyzed the differences in the protein cargo of EVs isolated from breast cancer cell lines and the plasma of breast cancer patients and healthy donors to identify the structural features that contribute to altered functional activities [171]. EVs from peripheral plasma samples of patients presented an increased degree of invasion of non-invasive breast cancer cells. Breast cancer plasma EVs displayed a unique proteome profile, with typical EV markers, such as CD9, CD81, CD63, and HSP70; Rab proteins; and clathrin. Moreover, the data were used in a comparison between the gene expression of MCF-10 treated with EVs obtained from breast cancer plasma patients, healthy donors, MDA-MB-231 cells, and untreated cells. The results displayed altered gene expression associated with angiogenesis, cell adhesion, and proliferation. In particular, it was confirmed that EVs released from aggressive breast cancer cells could stimulate the invasive behavior of non-invasive breast cancer cells. Moreover, the EV content from breast cancer patients can be used as a biomarker once these EVs contain a specific set of proteins [171].

### 3.2. Protein Markers in EVs Associated with Breast Cancer Features

Cancer-delivered EVs are a potential source of markers due to the fact that the cargo of EVs is enriched with molecules associated with tumor progression, metastasis, and invasion [199,200,201]. Rontogianni et al., analyzed the proteome content of EVs to discriminate cancer types and subtypes [176]. For this analysis, the authors used ten BC cell lines to identify a protein signature in EVs for use as a biomarker for breast cancer monitoring, diagnosis, and subtyping. Among the 4676 identified proteins, 14-3-3 proteins, integrins, annexin proteins, and cytoskeletal proteins, which are required for intermediate EV formation, were identified in all ten cancer cell lines. To discriminate the breast cancer subtypes, a total of 64 proteins were differentially expressed in TNBC compared to HER2+; these proteins are involved in angiogenesis (PLAU, ADAM9, and EPHA2), integrin-binding (ITGA5 and TIMP2), and cell motility (VIM and AXL). HER2+ EVs presented proteins associated with translation (EIFs), axon guidance (DNM2 and PIK3R1), and ERBB signaling (GRB7 and SHC1). Periostin, an extracellular matrix component, was described as a metastatic breast cancer biomarker identified in exosomes. It was observed that this protein was up-regulated in both murine and human cancer cell lines. Periostin has been reported to be highly expressed in several cancer types, including breast cancer [202], and it has been associated with osteoblast adhesion and cancer cell migration and is also involved in tumor angiogenesis [203]. Increased abundance of the Transient Receptor Potential Channel 5 (TrpC5) in EVs was observed in adriamycin-resistant human breast cancer cells (MCF-7/ADM) and was correlated with EV formation; additionally, adriamycin was found in EVs. The transference of TrpC5 to recipient cells mediated by EVs allows recipient cells to acquire TrpC5 and to P-glycoprotein multidrug transporter expression, conferring chemoresistance to non-resistance cells [204]. Considering the relevance of studies involving several BC cell lines, there are two main factors that must be considered. First, previous studies on tumor-derived EVs have determined that culture conditions can affect the omics components of EV cargo [205]. Second, despite the above, it is necessary to have a reliable collection of vesicular omics data from BC cell lines to find consistent markers for further research (Figure 1).

Regarding the use of liquid biopsies in relation to metastatic breast cancer, in one study, EV proteins were monitored to identify potential biomarkers for prognostics using a thermophoretic aptasensor. A total of 286 plasma samples from metastatic patients, non-metastatic patients, and healthy donors were submitted to a thermophoretic aptasensor to obtain a profile of eight EV markers (Figure 3): CA 15-3, CA 125, carcinoembryonic antigen (CEA), human epidermal growth factor receptor 2 (HER2), epidermal growth factor receptor (EGFR), prostate-specific membrane antigen (PSMA), epithelial cell adhesion molecule (EpCAM), and vascular endothelial growth factor (VEGF). These markers are involved in survival and migration, cell proliferation, metastasis, invasion, cell stemness, vascular permeability, and angiogenesis [200]. In an independent study, sera-derived exosomes from breast cancer patients were used to characterize CD24 and EpCAM as markers for exosomes (Figure 3). Western-blotting analysis was performed to identify exosomal markers in breast cancer samples. EpCAM and CD24 were identified in exosomes isolated from ascites ovarian cancer patients. However, only CD24 was detected in the EVs isolated from the serum of breast cancer patients. The absence of EpCAM in the serum-derived exosomes from breast cancer can be explained by the cleavage of EpCAM by serum metalloproteinases and the fact that both CD24 and EpCAM are not present in the same EV populations, which was confirmed in this study after the immunoaffinity purification of exosomes using anti-EpCAM beads [206]. Taken together, these studies show the importance of EV proteins as breast cancer markers for clinical use in order to diagnose, stratify, and monitor breast cancer progression.

### 3.3. EV Protein Glycosylation in Breast Cancer

Glycoconjugates such as glycoproteins functionally decorate the cell surface and are released in the extracellular milieu, playing an essential role in cell communication with the extracellular environment in several pathophysiological conditions, including cancer [207,208]. Any alteration in glycosylation levels due to the variation in the expression of glycosyltransferases and glycosidases [209], the dysregulation of chaperone activity [210], different levels of substrates, and alterations in nucleotide sugar transporter levels are diagnostic signs in cancer [211]. Aberrant glycosylation in tumor cells influences angiogenesis, cell proliferation, invasion, and metastatic processes [212,213,214,215]. The surface of an EV is enriched with glycoproteins, such as the Tetraspanin CD63, which is known to regulate cancer malignancy, and glycolipids [216]. EV glycosylation has been reported in several cancer types, including breast cancer. A list of 15 studies on EV glycosylation in breast cancer was reported by Macedo-Silva et al. (2021) [217], which is provided herein in Appendix A [169,170,172,182,183,184,185,190,218,219,220,221,222,223,224].

Nishida-Aoki et al. (2020) developed an EV glycosylation profile from breast cancer cells with brain metastasis tropism (BMD2a) comparing lymph node-metastatic (MDA-MB-231-luc-D3H2LN) and primary tumor-derived TNBC cell lines (MDA-MB-231-luc-D3H1) using lectin blotting [225]. The data showed a reduced level of sialic acid and a specific glycosylation pattern associated with galactose, GalNAc, lactose, and GalNAcα1−3GalNAc in BMD2a EVs. This study concluded that the glycosylation of EVs exerted an inhibitory effect on EV uptake [225]. This evidence shows that EV uptake depends on glycosylation profiles, as the authors showed that the cessation of O-glycosylation increases tumor EV accumulation in the lungs; meanwhile, N-glycosylation inhibition does not alter tumor EV biodistribution. It appears that the specific glycosylation profile in tumor EVs highlights their importance in breast cancer aggressiveness. For instance, EVs secreted by TNBC cells present higher levels of sialylation when compared to parental cells [226], and this process could interfere with EVs uptake. Sialic acid is negatively charged; its removal from the surfaces of EVs can promote the interaction of EVs and recipient cells through the uncovering of carbohydrate ligands that bind to lectins and promote the uptake of EV by recipient cells. Additionally, EV cargo contains glycosylated proteins that can interfere with EV–cell interaction. Cytokines binding to surface glycosaminoglycan side chains of proteoglycans decorate EVs isolated from breast cancer patients, altering EV biodistribution and leading to a higher metastatic burden [227]. The structure of bisecting GlcNAc glycan (β1,4-linked GlcNAc at the β-mannose residue core) on target proteins (EGFR and integrins) plays a role in cell metastasis and adhesion. The role of bisecting GlcNAc of EVs in breast cancer cells was described by Tan et al. (2020) [228]. Some proteins were described as being markers of EVs from tumor cells in breast cancer, such as extracellular matrix metalloproteinase inducer (EMMPRIN). A higher abundance of glycosylated EMMPRIN was identified in EVs from metastatic breast cancer patients. PNGase F treatment resulted in EVs’ deglycosylation and inhibited EV-induced invasion [222]. Somehow, glycans being components of EV cargo seems to dictate cell function. ST6Gal1-mediated sialylation modulates cell surface receptor function, promoting cell proliferation and invasion. Hait et al., showed that ST6Gal1 RNA transcripts and protein expression were variable and heterogeneous in a panel of TNBC and ER+ cell lines. However, EVs containing ST6Gal1 potentiate aggressive cancer cell growth, proliferation, and invasion in cells containing low amounts of endogenous ST6Gal1 [229]. In a heterogeneous tumor microenvironment, EVs act as important mediators of tumor progression, dictating EV organ tropism and metastasis. CA15-3/CA27.29 epitopes from Mucin 1 were reported to be tumor markers for breast cancer diagnosis. Additionally, alterations in Mucin 1 glycosylation profiles were observed in several cancer types, including human breast cancer, and in exosomes derived from luminal A breast carcinoma cells MCF-7 [170,230]. Extracellular matrix protein nephronectin (NPNT) could be present in a truncated or a glycosylated form in EVs secreted by mouse breast carcinoma cells [231]. N-glycosylation sites in integrin β1 modulate cell migration and the adhesion of MCF7 cells (luminal A), activating focal adhesion kinase (FAK) signaling, and these sites are crucial to integrin activity in EVs [232]. Additionally, exosomes released by MCF-7 cells after irradiation (luminal A type) increase concentrations of the enzyme GalNAc-T6, which is important for promoting epithelial–mesenchymal transition [233]. Furthermore, there is still a gap concerning whether alterations in EV glycosylation are due to changes in glycoprotein levels or changes in the expression/activity levels of glycan-modifying enzymes (Figure 1).

### 3.4. EV Proteome during the Metastatic Process in Breast Cancer

A screening of the literature regarding EV proteins identified in human and other species’ breast cancer cells, tissues, and biofluids, focusing on the TNBC and metastatic cases, allowed us to retrieve 22 studies and a total of 7265 proteins [14,170,171,173,175,176,177,178,179,181,184,186,187,188,221,234,235,236,237,238,239,240] (Appendix A). Interestingly, we observed that one set of proteins was identified more frequently (Figure 4A), with HLA-A, HLA-B, A2M, ACTB, ALDOA, ANXA2, and FN1 identified in at least 13 of 22 studies that explored the vesicular proteome in breast cancer.

Among the most frequent proteins, those located in membranes (plasma and organellar) and the cytoskeleton were particularly prominent (Figure 4A). On the other hand, after evaluating the subcellular localization of all the proteins identified at least twice in the EVs derived from breast cancer cells, it was determined that the proteasome complex, cell–cell adhesion junctions, focal adhesions, extracellular matrix, extracellular exosome, and membranes were also relevant (Figure 4B). It is possible to identify vesicular proteins that are involved in different phases of the cell cycle and the p53-linked response (Figure 4C), reinforcing the role of EVs in metastatic processes and breast cancer progression.

In fact, studies have demonstrated the crucial role of EVs in pro-metastatic processes, wherein a cancer cell’s communication with its local environment is necessary to initiate growth and invasion through the transfer of molecules, which include mRNAs, microRNAs, and proteins [235]. Different studies have revealed the vesicular proteins associated with metastatic events (Table 1), and among those listed, ADAM10 (four studies), VIM (three studies), and ANXA6 (two studies) stand out. An important finding was that exosomes expressing these proteins play important roles in TNBC (Figure 3).

It has been identified that the levels of the CD151 protein are increased in serum exosomes from TNBC patients, and this protein has been shown to promote proliferation, migration, and invasion by modulating the activities of laminin-binding integrins α3β1, α6β4, and α6β1 [175]. An increase in CD151 abundance was also verified in a TNBC cell model (MDA-MB-231) [181]. These results indicate that exosomal CD151 can be used as a biomarker to distinguish TNBC patients from healthy individuals.

The data gathered in Table 1 reveal the importance of other vesicular proteins in TNBC, such as VIM (three studies) and ANXA6, EDIL3, EPHA2, FLNA, FLNB, and PLAU (two studies). It has been observed that the VIM protein and its gene play important roles in controlling migration and invasion in different types of cancer [242,243]; here, it is highlighted that its location in EVs can be associated with the aggressiveness of cancer. Based on the collected data, it is possible to select possible markers of EVs in breast cancer. In addition to conventional markers such as CD9, CD81, and CD83, the proteins HLA-A, HLA-B, HSPA5, A2M, ACTB, ALDOA, ANXA2, FN1, ANXA5, GSN, MYH9, PGK1, and ACTN1 are also often identified in EVs in cells and breast cancer patients at different stages. Therefore, these markers could be used to purify EVs potentially related to highly aggressive tumor subpopulations and study them separately (Figure 1).

## 4. Current Status of BC-Related Biomarkers in EVs

EVs are natural sources of cancer biomarkers. In Table 2, we summarize potential markers described as vesicular cargo.

Considering that miRNAs are the most-studied components in the vesicular context, most putative markers for BC-related EVs depend on these non-coding RNA sequences. Herein, we found potential biomarkers for diagnosis, classification, prognosis, and the prediction of treatment response in relation to BC. Despite differences in the methods used for EV isolation and target selection, all area-under-the-curve (AUC) values shown for the diagnosis of BC were higher than 0.7.

Notably, vesicular hsa-miR-21 was identified as a relevant candidate for diagnosing BC patients in different cohorts [38,39,106]. According to some studies, its performance could be improved by estimating multi-miRNA scores [106] or by combining the levels of this miRNA with MMP1/CD63 protein quantification [37]. In addition, vesicular hsa-miR-421 levels demonstrated the potential to discriminate BC patients from healthy individuals, even those with early stages of the disease [115]. Furthermore, the analysis of vesicular miRNA levels revealed a dual function of some miRNAs. For example, hsa-miR-1246 [38,98] and hsa-miR-142-5p [244] allow for the diagnosis of BC patients and contribute to estimating the treatment response or classifying BC patients, respectively.

Moreover, other kinds of RNA markers (lncRNA, mRNAs, and circRNA) are currently being studied to characterize their features in EVs from BC patients (Table 2) and accumulate evidence to improve current classification systems. It is important to note that classification features can be enhanced if we specially analyze tumor-derived EVs. Although we still lack an exclusive tumor-cell-derived BC marker for EVs, some tumor markers, such as CD49f or the epithelial cell-specific marker (EpCAM), have shown promising results in this regard [106].

## 5. EVs in Oncological Precision Medicine and Treatment: Promising Avenues and Open Questions

There is no doubt that EVs are an important component of the tumor microenvironment, and a great deal of advancement has been made in understanding how these subcellular nanocompartments affect tumor progression and dissemination. Since tumor-derived EVs can be found in all biological fluids and EV cargo contains genetic information from parental cells, they have been considered potential candidates for liquid biopsies. One approach for specifically investigating circulating EVs derived from tumor cells is based on tumor surface markers that are present in EV membranes. In this context, glycoconjugates have recently attracted attention. Specific antibodies or lectins that recognize the aberrant glycosylation patterns of tumors serve as potential tools that can be used to isolate circulating-tumor-derived-EVs, allowing for the investigation of vesicular components such as RNA and proteins with clinical significance as biomarkers. 

Nevertheless, despite EVs’ promising applicability and the advances in their use as a liquid biopsy tool, there are still some pitfalls that should be addressed by researchers. Technical challenges concerning effective and selective EV isolation, as well as EV nomenclature, still require better standardization to allow for the comparison of data between different groups. Additionally, we should certainly consider additional factors or biases that affect the collection of EVs from biological fluids like handling, storage, the use of anticoagulants, the volume of blood drawn, age, sex, ancestry groups, disease state (in addition to other comorbidities), medication, circadian rhythm, body mass index, hydration, and dietary status. Indeed, all these issues should be considered when data are interpreted. In addition, another layer of complexity can be added with respect to the methods for the quantitative analysis of EV cargo. First, the quantification of small amounts of RNA and, to a certain extent, protein are non-trivial. In particular, the required input of RNA often does not match the minimal amount recommended for different downstream procedures, leading to a poor signal-to-background ratio. Furthermore, the lack of normalization methods for the quantitative analysis of nucleic acids, proteins, and other biological molecules constitutes yet another gap in the field. Therefore, the scientific community is engaging in extensive discussions and research to establish pipelines to fill these gaps.

Despite the many issues mentioned above, better biomarkers for early detection and predicting therapeutic response are still needed in order to provide a more favorable clinical outcome. As for any new, growing field in science, some technical obstacles, such as those mentioned above, are still a reality; however, due to the number of circulating EVs and their nature as intrinsic sources of biological molecules for modulating cell phenotypes, we believe that circulating EVs hold a great deal of potential in the clinical setting since they represent a method for tracking cancer-associated changes in a personalized manner. Moreover, as the field advances toward overcoming these limitations, the information gleaned from EVs will certainly represent an interesting approach not only to understanding the clinical evolution of BC but also to revealing new routes for clinical intervention.

From the perspective of therapy, EVs have been considered as an important tool with which to improve cancer treatment. Since EVs are natural carriers, some strategies focus on loading EVs with therapeutic components such as chemotherapeutic drugs, siRNAs, peptides, and proteins, presenting significantly enhanced anti-tumor effects [251]. Different methods can be used for EV loading, including electroporation, extrusion, sonication, freeze–thawing, the use of pH gradients, ultrasound, the use of streptomycin O, and hybridization. Another alternative that has been used is the anchorage of a drug on an EV’s surface, which can be achieved via click chemistry, to modify the surface of EVs (reviewed in [252]).

Another approach of EV engineering is more indirect and involves the manipulation of the cellular EV source. In this approach, producing cells are incubated with therapeutic agents to increase the loading of these drugs into EVs. Furthermore, the producing cells can be transfected or infected with plasmids or viral vectors to overexpress specific molecules in secreted EVs. The presence of these biomolecules on an EV’s surface can be a promising strategy for tumor targeting and specific delivery, thereby improving therapeutic efficacy and reducing toxicity [253]. Nevertheless, it should be mentioned that there is not a gold-standard method for modifying EVs, and each one has its own advantages and disadvantages. Nonetheless, there are a few ongoing clinical trials using engineered EVs for cancer treatment (Clinical Trials, NCT03608631 and NCT01294072).

## 6. Conclusions

EVs play a substantial role in BC initiation and progression, impacting the natural history of the disease. Tumor-derived EVs are biological nanostructures that carry important genetic information from tumor cells and can be found in all biological fluids, representing a natural source of diagnostic and prognostic biomarkers in cancer. Within this context, finding a set of biomarkers in circulating EVs able to (1) discriminate between healthy individuals and BC patients, (2) discriminate between different molecular subtypes of BC, and (3) determine the probability of disease recurrence would represent a game changer in the field that could pave the way for precision oncology medicine. In addition, in the therapeutic setting, EV engineering can greatly improve the specificity and effectiveness of tumor therapy, as demonstrated in pre-clinical studies. However, further efforts are still necessary to better standardize technical procedures for EVs’ isolation and cargo identification along with the establishment of reliable assays with which to evaluate the therapeutic potential of EVs in clinical practice.

## Figures and Tables

**Figure 1 ijms-24-13022-f001:**
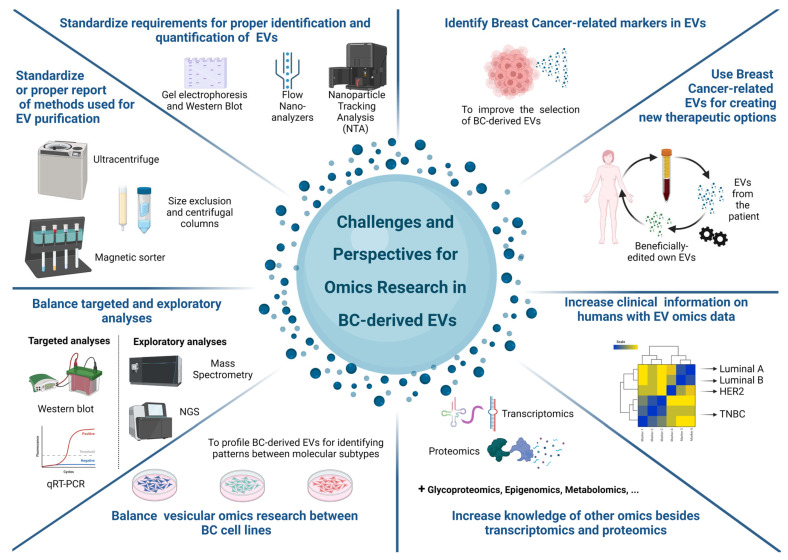
Challenges and perspectives regarding omics research on BC-derived EVs. This review presents the current state of the art of the most-studied omics topics on EVs from BC, namely, transcriptomics and proteomics. However, we must fill omics-related gaps before proposing reliable EV-based tools for this disease. Here, we cited some challenges for future research. As vesicles are heterogeneous in terms of size, biogenesis, and cargo, authors must standardize the reporting of methods for the isolation, quantification, characterization, and profiling of EVs. Furthermore, consistent findings in relation to EVs are characterized by their ability to be replicated. Nevertheless, many studies use targeted analysis approaches, which can bias observations. In addition, such replicability must be related to characterizing different individuals of the same subgroup or cell lines of the same subtype. To evaluate this correctly, it is necessary to increase the number of studies comparing less-studied BC cell lines and include a translational approach between tumor cell markers and their vesicular pairs. Regarding associations with the subtype of BC patients, there are gaps produced by the lack of available information about the molecular or clinical profiles of these patients, which can complicate future secondary analysis. After conquering this challenge, we can promisingly combine data from different omics studies of BC-derived EVs and select potentially tumor-derived EVs via liquid biopsies from patients to debug or edit these vesicles and induce a beneficial effect in BC patients. Image created on BioRender.com.

**Figure 2 ijms-24-13022-f002:**
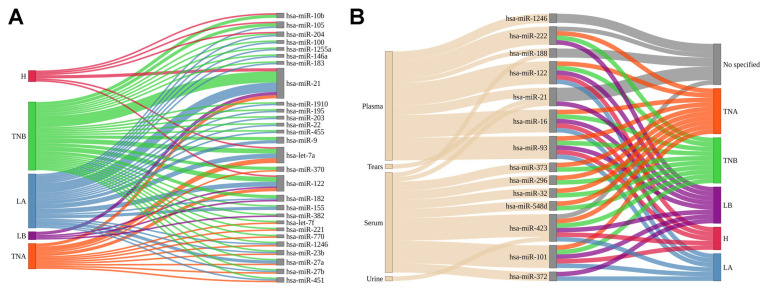
Extracellular vesicle (EV) miRNAs in breast-cancer-related studies. Sankey plots show the number of studies mentioning each relevant vesicular miRNA from cell supernatant (**A**) or human bodily fluids (**B**). The cell lines in which the EV cargo was analyzed are classified into the main BC subtypes following the criteria given in Dai et al.’s (2017) study [124]. For studies on EVs collected from BC patients, the subtype information was retrieved from each study. H: Her2, TNA: Triple-Negative A, TNB: Triple-Negative B, LA: Luminal A, and LB: Luminal B.

**Figure 3 ijms-24-13022-f003:**
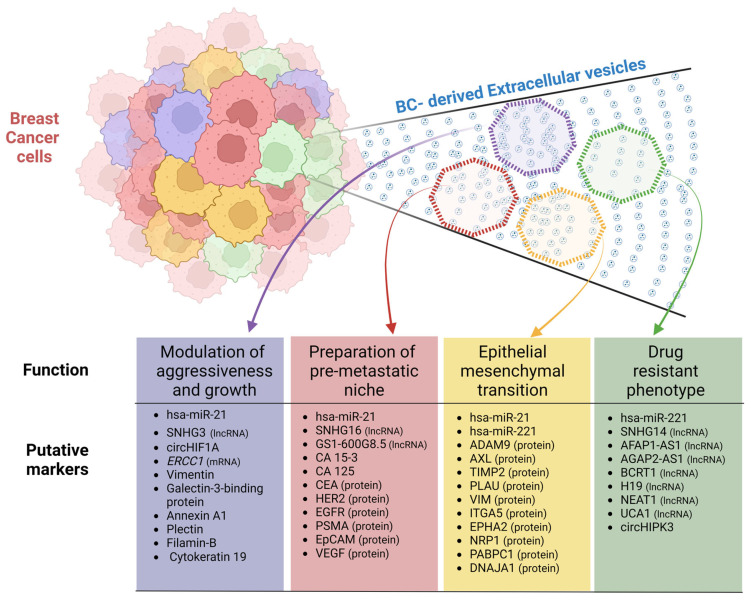
Relevant putative markers in BC-derived EVs. Breast cancer cells produce a great diversity of EVs. These EVs can be classified into subpopulations based on their proteomic and transcriptomic cargo. In this review, we associate some BC-derived EV subpopulations with tumor-related functions. In addition, we include putative markers related to their types (miRNA, lncRNA, mRNA, circRNA, or protein) for each subpopulation. Image created on BioRender.com.

**Figure 4 ijms-24-13022-f004:**
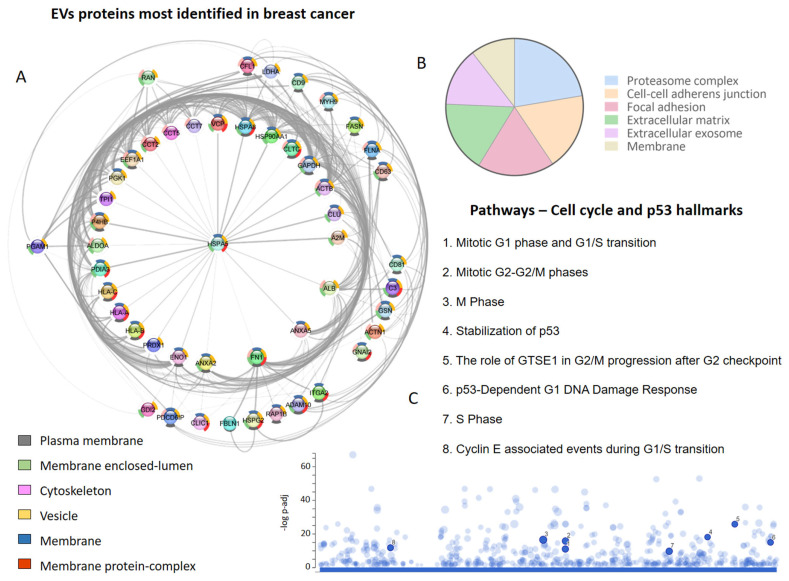
Extracellular vesicle (EVs) proteins in breast cancer proteome studies. (**A**) The most frequently identified proteins in the evaluated studies. The donut graph shows the corresponding subcellular locations. (**B**) Subcellular localization of proteins identified in EVs in at least two studies. (**C**) Cell-cycle-related and p53 pathways in which EV proteins participate.

**Table 1 ijms-24-13022-t001:** Protein clusters found in EVs derived from cell lines and bodily fluids from humans with breast cancer.

Protein Clusters	Highlight	BC-Related Subtype ^1^ (Cell Lines) or Type of Sample (Bodily Fluids)	Reference
PDIA1, PDIA3, SUSD2, LGALS3BP, PRKCSH	The SKBR3 cell line presents a higher number of peptide spectral matches with invasion-related proteins compared to MCF7 and MCF10.	Her2 and Luminal A	[186]
FZD6, DVL1, PK1, VANGL1	Inhibition of Fzd6, Dvl1, Pk1, and Vangl1 (PCP pathway) reduced tumor cell motility.	Triple-Negative B	[238]
VIM, LGALS3BP, ANXA1, EDIL3, FLNB, TGM2, CTSD, PLAU, PRSS23, SERPINE1, CLIC1, EPHA2, NDRG1, HSPA9, HSP90AB1, SRP68	There are relatively few proteins that are differentially expressed in exosomes from MDA-MB-231 cells (invasive/metastatic) and MCF-7 cells (non-invasive/non-metastatic). However, the identified proteins that were upregulated at MDA-MB-231 have known functions in migration.	Luminal A and Triple-Negative B	[14]
C4B, EDIL3, VTN, ANXA6, NUMA1, FLNC, TXNRD1, COL5A1, PSMB4, FLNA, EEA1, ENPP1, PSMA1, ILF2, RPL12, ISG15, PSMB7, PON1, PSMD6, LAMP1, FTL, PSMC5, RPL27	EVs from young women breast cancer patients drive increased invasion of non-malignant cells via the Focal Adhesion Kinase pathway. These results suggest that the protein content of EVs from MDA-MB231 and MCF10DCIS.com cells reflects the biologic differences between these invasive and non-invasive breast cancer cells.	Triple-Negative B	[171]
ZEB1, SNAI1	EVs from metastatic cells can affect the behavior of less-aggressive neighboring cells.	Luminal A and Triple-Negative B	[237]
ADAM10	Exosomal ADAM10 increases aldehyde dehydrogenase expression in breast cancer cells through Notch receptor activation and enhances motility through the GTPase RhoA.	Triple-Negative B	[239]
ADAM10	ADAM-10 expression levels are increased in exosomes from luminal cancer subtype blood in comparison with healthy patients.	Blood	[173]
PEPD, NCL, PARP1, ACTA2, ACTG2, TBCA, TTYH3, MATR3, KPNB1, KRT16, RANBP2, CCT6A	The application of protein signatures to discriminate breast cancer patients with or without metastasis yielded a sensitivity of 81% and a specificity of 81%.	Serum	[241]
CD151	CD151 levels in serum exosomes derived from TNBC were significantly higher than the levels of exosomes from healthy individuals. This protein facilitates the secretion of ribosomal proteins via exosomes while inhibiting the secretion of complement proteins. CD151-deleted exosomes significantly decreased cell migration and invasion.	Serum	[175]
ADAM9, AXL, TIMP2, PLAU, VIM, ITGA5, EPHA2, NRP1, PABPC1, DNAJA1	TNBC-specific signature proteins featured prominently in angiogenesis, cell motility and cell migration, and integrin binding.	All main subtypes	[176]
A1BG, ACSM3, ADAM10, AHSG, AMBP, APOA1, APPBP2, BANF1, BMP7, C3, CLK3, ELL2, GTSE1, HIG1AN, HMOX1, IGF2R, ITIH4, KRT1, KRT6A, KRT6B, MBD4, PCNT, PIBF1, PPM1A, SERPINA1, SERPINB7	Exosomal proteins from the blood of breast cancer patients are associated with crucial steps of tumor progression and metastasis.	Blood	[177]
ANXA5, ANXA6, VIM, CD44, EIF4A1, ITGA2, CD147, ENO1, ITGB1, FLNB, FLNA, EGFR	It was observed that the metastatic proteins ENO1 and ITGB3 were upregulated in all biological replicates of MDA-MB-231 EVs.	Luminal A and Triple-Negative B	[179]
GLUT-1, ADAM10, GPC-1	GPC-1, ADAM10, and GLUT-1 proteins may be novel potential biomarkers for breast cancer detection and prognosis.	Luminal A and Triple-Negative B	[181]

^1^ Cell lines were classified into subtypes using the data reported in study conducted by Dai et al. (2017) [124].

**Table 2 ijms-24-13022-t002:** Features of vesicular cargo conducive to their use as potential biomarkers for breast cancer.

Markers	RNA Type	Marker Type	Comparison (Group 1 vs. Group 2)	Source	Method for EV Isolation	Total or Specific Group of EVs	Method for Vesicular RNA Isolation	Method for Vesicular RNA Quantification	Area under the Curve	Ref.
hsa-let-7a-5p and hsa-miR-222-3p	miRNA	Classification	Inflammatory BC (*n* = 23) vs Non-Inflammatory BC (*n* = 34)	Plasma	Combination of methods	Total	Organic extraction method	qRT-PCR	0.973	[100]
hsa-miR-142-5p	miRNA	Classification	TNBC (*n* = 15) vs. Luminal A (*n* = 16)	Serum	Precipitation using chemical reagent	Total	Organic extraction method	qRT-PCR	0.921	[244]
hsa-miR-4448, hsa-miR-2392, hsa-miR-2467-3p and hsa-miR-4800-3p	miRNA	Classification	recurrent TNBC (*n* = 12) vs. Non recurrent TNBC (*n* = 12)	Serum	Precipitation using chemical reagent	Total	Organic extraction method	qRT-PCR	0.765	[245]
*HLA-DRB1, HAVCR1, ENPEP, TIMP1, CD36, MARCKS, DAB2,* and *CXCL14*	mRNA	Diagnosis	BC patients (*n* = 101) vs. Healthy individuals or those with benign breast disease (*n* = 81)	Plasma	Affinity-based binding to spin columns	Total	Organic extraction method	qRT-PCR	0.77	[246]
*PGR, ERBB2*	mRNA	Diagnosis	BC patients (*n* = 63) vs. Healthy individuals (*n* = 20)	Plasma	Ultracentrifugation	Total	Affinity-based binding to spin columns	ddPCR	0.93	[35]
hsa-miR-424, hsa-miR-423, hsa-miR-660, and hsa-let-7i	miRNA	Diagnosis	BC patients (*n* = 69) vs. Healthy individuals (*n* = 40)	Urine	Size-exclusion filter	Total	Silicon-carbide-based method	qRT-PCR	0.995	[119]
hsa-miR-21	miRNA	Diagnosis	BC patients (*n* = 30) vs. Healthy individuals or other tumors (*n* = 54)	Plasma	Ultracentrifugation	Total	Organic extraction method	qRT-PCR	0.961	[39]
hsa-miR-16, hsa-miR-21, hsa-miR-429, and hsa-miR-9	miRNA	Diagnosis	BC patients (*n* = 62) vs. Healthy individuals (*n* = 20)	Plasma	Microfluidic device	CD49f and EpCAM-positive EVs	Organic extraction method	qRT-PCR	0.880	[106]
hsa-miR-142-5p, hsa-miR-320a, and hsa-miR-4433b-5p	miRNA	Diagnosis	BC patients (*n* = 31) vs. Healthy individuals (*n* = 16)	Serum	Precipitation using chemical reagent	Total	Organic extraction method	qRT-PCR	0.839	[244]
hsa-miR-421	miRNA	Diagnosis	BC patients (*n* = 20) vs. Healthy individuals (*n* = 10)	Plasma	Ultracentrifugation	Total	Organic extraction method	qRT-PCR	0.835	[115]
hsa-miR-421	miRNA	Diagnosis	stage I BC patients (*n* = 13) vs. Healthy individuals (*n* = 10)	Plasma	Ultracentrifugation	Total	Organic extraction method	qRT-PCR	0.809	[115]
hsa-miR-375, hsa-miR-655-3p, hsa-miR-548b-5p, and hsa-miR-24-2-5p	miRNA	Diagnosis	stage I BC patients (*n* = 12) vs. Healthy individuals (*n* = 10)	Plasma	Precipitation using chemical reagent	Total	Organic extraction method	RNA-seq	0.808	[104]
hsa-miR-17-5p	miRNA	Diagnosis	BC patients (*n* = 83) vs. Healthy individuals (*n* = 34)	Serum	NA	NA	NA	qRT-PCR	0.784	[247]
hsa-miR-1246 and hsa-miR-21	miRNA	Diagnosis	BC patients (*n* = 16) vs. Healthy individuals (*n* = 16)	Plasma	Combination of methods	Total	Organic extraction method	RNA-seq	0.727	[38]
BEX2, AC104843.1, AL136981.2, KRT19, NPM1P25, CTSG, CBR3, HOXB7, AL691447.3, RNA5SP141, and circRNA chr13_42953948_42970670_-	lncRNA /circRNA	Diagnosis	stages I-II BC patients (*n* = 63) vs. Healthy individuals or those with benign breast disease (*n* = 60)	Plasma	Affinity-based binding to spin columns	Total	Organic extraction method	RNA-seq	0.940	[248]
HOTAIR	lncRNA	Diagnosis	BC patients (*n* = 15) vs. Healthy individuals (*n* = 15)	Serum	Combination of methods	Total	Affinity-based binding to spin columns	qRT-PCR	0.918	[249]
H19	lncRNA	Diagnosis	BC patients (*n* = 50) vs. Healthy individuals or those with benign breast disease (*n* = 100)	Serum	Precipitation using chemical reagent	Total	Organic extraction method	qRT-PCR	0.870	[250]
hsa-miR-150-5p	miRNA	Prognosis	Recurrent BC (*n* = 12) vs. non-recurrent BC (*n* = 15)	Plasma	Precipitation using chemical reagent	Total	Organic extraction method	RNA-seq	0.705	[132]
hsa-miR-1246 and hsa-miR-155	miRNA	Treatment Response Prediction	Tratuzumab-resistant (*n* = 32) vs. Trastuzumab-sensitive BC patients (*n* = 36)	Plasma	Precipitation using chemical reagent	Total	Magnetic-beads-based method	qRT-PCR	0.898	[98]
MSMO1	lncRNA	Treatment Response Prediction	Patients with PCR (*n* = 24) vs patients with no-PCR after treatment with NACT (*n* = 34)	Plasma	Affinity-based binding to spin columns	Total	Organic extraction method	RNA-seq	0.79	[248]

Ref.: Reference; BC: Breast Cancer; TNBC: Triple-Negative Breast Cancer; EV: Extracellular Vesicle; PCR: pathological complete response; NACT: neoadjuvant chemotherapy; NA: Not Available.

## Data Availability

No new data were created or analyzed in this study. Data sharing is not applicable to this article.

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
