# Peer review of "Deciphering the Functional Status of Breast Cancers through the Analysis of Their Extracellular Vesicles"

_ijms, 2023, doi:10.3390/ijms241613022_

Round 1
Reviewer 1 Report
The review comments for the manuscript, ' Deciphering the functional status of breast cancers through the analysis of their extracellular vesicles', are given below,
1. The work is impressive and presented well. Still, few changes need to be done for betterment.
2. Quality of all the figures must be improved.
3. Abstract and conclusion must be given brief and precise.
4. Adding nomenclature for the symbols and abbreviations used in the manuscript may help the readers.
5. Also Introduction section need to be updated with more recent references, I suggest the following, 10.3390/life12030426
6. References should not be grouped like, [17, 26–28] or [29,31,32,41–116]. Each and every reference must be properly justified and cited.
7. Add the organization of the manuscript at the end of Introduction section.
Authors need to check the manuscript throughout for typos and grammatical errors. English must be improved with the help of a language expert.
Author Response
Reviewer #1
Point 1.1:
The review comments for the manuscript, ' Deciphering the functional status of breast cancers through the analysis of their extracellular vesicles', are given below,
- The work is impressive and presented well. Still, few changes need to be done for betterment.
Response 1.1:
Thank you for your kind comment about our study. We have revised all reviewer’s comments for their implementation to improve our manuscript.
Point 1.2:
- Quality of all the figures must be improved.
Response 1.2:
We apologize we lost figure resolution when we pasted figures on the Word document. In this submission, we attached all figures as supplementary material to see them individually.
Point 1.3:
- Abstract and conclusion must be given brief and precise.
Response 1.3:
Thank you for pointing this out. We revised the text to be more concise and reduce potential redundant clauses, especially in the abstract and conclusion.
Point 1.4:
- Adding nomenclature for the symbols and abbreviations used in the manuscript may help the readers.
Response 1.4:
Thank you for commenting on this. We followed the journal guidelines (https://www.mdpi.com/journal/ijms/instructions) to define all Acronyms/Abbreviations/Initialisms in their first mention in the abstract, the main text, or the first figure/table.
Point 1.5:
- Also Introduction section need to be updated with more recent references, I suggest the following, 10.3390/life12030426
Response 1.5:
Thank you for this suggestion. We have revised the mentioned manuscript. Nevertheless, this research focuses on different topics beyond breast cancer, and/or extracellular vesicles. Unfortunately, we didn’t find a way to include this paper in our review.
Point 1.6:
- References should not be grouped like, [17, 26–28] or [29,31,32,41–116]. Each and every reference must be properly justified and cited.
Response 1.6:
Thank you for pointing this out. We understood your comment about references, but as extracellular vesicles in cancer are a relatively new field of knowledge, we aim to present information consistently reported in many studies. In addition, some studies, according to their content, were referenced many times in different topics of our review. It represents one strength of our review once we included more than 260 references. Subsequently, we followed the IJMS author’s guidelines for formatting them.
Point 1.7:
- Add the organization of the manuscript at the end of Introduction section.
Response 1.7:
Thank you for this comment. We apologize for the unnecessary spaces in the organization of topics (between sections 2.1 and 2.2). We revised the manuscript structure and followed the IJMS author’s guidelines. If the manuscript is accepted for publication, the online version of the published manuscript will include the manuscript organization in a lateral panel.
Reviewer 2 Report
Nice topic. However there is some work to be added before recommending this for publication.
1) Please try to shorten the review since there is a lot of things explained and repeated several times which is not needed
2) Write more about EVs metabolic features in BC vs other cancers
3) Try to correlate what is found up to date and make a suggestion about potential biomarkers this would be great
4) Talk more about population differences
No comments
Author Response
Reviewer #2
Point 2.1:
Nice topic. However there is some work to be added before recommending this for publication.
1) Please try to shorten the review since there is a lot of things explained and repeated several times which is not needed
Response 2.1:
Thank you for pointing this out. We revised the text to be more concise and reduced potential redundant topics.
Point 2.2:
2) Write more about EVs metabolic features in BC vs other cancers
Response 2.2:
Thank you for this comment. In this version, we revised information about extracellular vesicle metabolic features in breast cancer in lines. As this topic is still a growing field, we included functional studies based on metabolic effects regarding nucleic acids or protein cargoes as follows:
- “The functions of EVs proteins from SKBR3 are related to metabolism (Enolase, Fatty acid synthase, Phosphoglycerate kinase, Fructose bisphosphatase 1, GAPDH, Malate dehydrogenase, L-lactate dehydrogenase, Aldehyde dehydrogenase, Aldolase, Triosephosphate isomerase, and Glucosidase 2 subunit beta), binding (Selenium-binding protein 1, 60 kDa heat shock protein, Protein disulfide-isomerase, Lamin A/C, Tumor protein D52), and assembly (Myosin-9, alpha-Actinin-4, Cytokeratin 16, Cytokeratin 18, Cytokeratin 8, Cytokeratin 19). Although MCF-7 EVs protein content displays the same molecular function as SKBR3, different proteins were identified: Aldolase, Pyruvate kinase, Tryptophan-tRNA ligase, Cathepsin D, Kynureninase, TER ATPase, Lactoferroxin-C, and Hexokinase-1 which are involved in metabolism, and HSP90-a, Agrin, and protein SET are related to binding.” (lines 509-519)
- “In the same way that vesicular lncRNA from BC cells can modify the metabolism of other cell types, EVs produced in other cell types affect tumor growth. As an example, BC-modulating vesicular HISLA and SNHG3 lncRNAs are expressed from tumor-associated macrophages (TAM) and cancer-associated fibroblasts (CAF), respectively. HISLA avoids the interaction between PHD2 and HIF-1α for accumulating HIF-1α, which regulates the glycolytic metabolism in the tumor cell [153]. SNHG3 sponges hsa-miR-330-5p in tumor cells regulating PKM positively, which increases tumor growth [154]. However, transcriptome from BC-derived EVs also includes suppressor lncRNAs such as XIST, whose repression promotes brain metastasis in BC patients [155]. The absence of XIST induces an accumulation of hsa-miR-503, which triggers M1-M2 polarization for suppressing T-cell proliferation allowing tumor expansion” (lines 315-325).
- “The current development of analytical methods, together with the optimization of pre-analytical factors and computational strategies, allowed a better understanding of the EVs lipid composition [193]. A comparison of lipid composition between EVs and cells of origin [194] has shown that in vitro, EVs have enrichment in cholesterol, sphingomyelin, glycosphingolipids, and phosphatidylserine compared to their origin cells while phosphatidylcholine and phosphatidylinositol are enriched in cells compared to exosomes. Quantitative analyses of oxysterols in exosomes released from breast cancer cells revealed that levels of 27-Hydroxycholesterol were higher in exosomes from MCF-7 cells compared to MDA-MB-231 and non-cancerous cells, showing a dependence of the levels of this exo-somal lipid on the ER status of the cell of origin [195]. Another study compared the lipid profile of EVs isolated from TNBC cell lines, D3H2LN and D3H1, with high and low metastatic potential, respectively [196]. The exosomal levels of unsaturated diacylglycerols isolated from high metastatic cells were higher and stimulated angiogenesis through protein kinase D signaling pathway.” (lines 482-495).
Point 2.3:
3) Try to correlate what is found up to date and make a suggestion about potential biomarkers this would be great
Response 2.3:
Thank you for pointing this out. We included all potential biomarkers in Table 2 and Section 4. Current Status of BC-related Biomarkers in EVs. For being considered as biomarkers, we described all proposals whose specificity/sensitivity features were included in their original publications.
Point 2.4:
4) Talk more about population differences
Response 2.4:
Thank you for this comment. We explored specific studies which investigated extracellular vesicles in breast cancer patients from specific ancestry-related populations. Though many studies in this review involve patients from different ancestry groups, they have not included this factor in the analysis. In this version, we commented that our observations are based on results from patients studied in different countries and/or ancestry groups (lines 169-172). Also, we added the ancestry group as a potential source of bias in the collection/analysis of extracellular vesicles (New Section EVs in oncology precision medicine and treatment: promises and opening questions).
Reviewer 3 Report
The manuscript "Deciphering the functional status of breast cancers through the analysis of their extracellular vesicles" submitted makes a significant contribution to research wherein authors analyzed the cargoes of breast cancer-derived extracellular vesicles which yielded a list of potential markers. Authors have also discussed about the potential obstacles to the clinical application of EV-based tests and the technologies available for assessing their content to present a set of markers for further evaluation. This review is well-written, simple to understand for every reader that also “clearly” describes the purpose of this research. It's essential to bridge/address the gap, and the authors did a good job at it. Minor grammatical errors need to be corrected.
Minor grammatical errors need to be corrected.
Author Response
Reviewer #3
Point 3.1:
The manuscript "Deciphering the functional status of breast cancers through the analysis of their extracellular vesicles" submitted makes a significant contribution to research wherein authors analyzed the cargoes of breast cancer-derived extracellular vesicles which yielded a list of potential markers. Authors have also discussed about the potential obstacles to the clinical application of EV-based tests and the technologies available for assessing their content to present a set of markers for further evaluation. This review is well-written, simple to understand for every reader that also “clearly” describes the purpose of this research. It's essential to bridge/address the gap, and the authors did a good job at it. Minor grammatical errors need to be corrected.
Response 3.1:
Thank you for your kind comment and summary of our study. We revised the English grammar of our manuscript. We believe that the readability of this version has been improved.
Reviewer 4 Report
In the present study, Carrasco et al. comprehensively described the functional role of extracellular vesicles (EVs) in breast cancer. The manuscript is well-written, with lots of important information, and would be helpful for the researchers in the field. However, I would suggest some minor changes to improve that manuscript than its present form. The comments are shown below:
1. The authors should also mention the amphisomal mechanism of exosome biogenesis.
2. A couple of sentences about EVs' carried lipids and metabolites could be included.
3. Abstract: line 5; page 3, figure legend 1, line 8- please remove the extra spaces.
4. The authors should also discuss about the target specificity of EVs in therapeutic settings.
5. Please add the following references to acknowledge the role of EVs in breast cancer:
a). Ref. 1: Sci Rep. 2018 May 9;8(1):7357. doi: 10.1038/s41598-018-25725-w; PMID: 29743547
b). Ref. 2: Mol Carcinog. 2018 Dec;57(12):1707-1722. doi: 10.1002/mc.22891. Epub 2018 Sep 5; PMID: 30129687
Author Response
Reviewer #4
Point 4.1:
In the present study, Carrasco et al. comprehensively described the functional role of extracellular vesicles (EVs) in breast cancer. The manuscript is well-written, with lots of important information, and would be helpful for the researchers in the field. However, I would suggest some minor changes to improve that manuscript than its present form. The comments are shown below:
- The authors should also mention the amphisomal mechanism of exosome biogenesis.
Response 4.1:
Thank you for your kind comment and summary of our study. In this version, we added information about the amphisomal biogenesis of extracellular vesicles in lines 44-46.
Point 4.2:
- A couple of sentences about EVs' carried lipids and metabolites could be included.
Response 4.2:
Thank you for this comment. In this version, we revised information about extracellular vesicle metabolic/lipid features in breast cancer in lines. As this information is largely unknown, we included functional studies based on metabolic effects regarding nucleic acids or protein cargoes as follows:
- “The functions of EVs proteins from SKBR3 are related to metabolism (Enolase, Fatty acid synthase, Phosphoglycerate kinase, Fructose bisphosphatase 1, GAPDH, Malate dehydrogenase, L-lactate dehydrogenase, Aldehyde dehydrogenase, Aldolase, Triosephosphate isomerase, and Glucosidase 2 subunit beta), binding (Selenium-binding protein 1, 60 kDa heat shock protein, Protein disulfide-isomerase, Lamin A/C, Tumor protein D52), and assembly (Myosin-9, alpha-Actinin-4, Cytokeratin 16, Cytokeratin 18, Cytokeratin 8, Cytokeratin 19). Although MCF-7 EVs protein content displays the same molecular function as SKBR3, different proteins were identified: Aldolase, Pyruvate kinase, Tryptophan-tRNA ligase, Cathepsin D, Kynureninase, TER ATPase, Lactoferroxin-C, and Hexokinase-1 which are involved in metabolism, and HSP90-a, Agrin, and protein SET are related to binding.” (lines 509-519)
- “In the same way that vesicular lncRNA from BC cells can modify the metabolism of other cell types, EVs produced in other cell types affect tumor growth. As an example, BC-modulating vesicular HISLA and SNHG3 lncRNAs are expressed from tumor-associated macrophages (TAM) and cancer-associated fibroblasts (CAF), respectively. HISLA avoids the interaction between PHD2 and HIF-1α for accumulating HIF-1α, which regulates the glycolytic metabolism in the tumor cell [153]. SNHG3 sponges hsa-miR-330-5p in tumor cells regulating PKM positively, which increases tumor growth [154]. However, transcriptome from BC-derived EVs also includes suppressor lncRNAs such as XIST, whose repression promotes brain metastasis in BC patients [155]. The absence of XIST induces an accumulation of hsa-miR-503, which triggers M1-M2 polarization for suppressing T-cell proliferation allowing tumor expansion” (lines 315-325).
- “The current development of analytical methods, together with the optimization of pre-analytical factors and computational strategies, allowed a better understanding of the EVs lipid composition [193]. A comparison of lipid composition between EVs and cells of origin [194] has shown that in vitro, EVs have enrichment in cholesterol, sphingomyelin, glycosphingolipids, and phosphatidylserine compared to their origin cells while phosphatidylcholine and phosphatidylinositol are enriched in cells compared to exosomes. Quantitative analyses of oxysterols in exosomes released from breast cancer cells revealed that levels of 27-Hydroxycholesterol were higher in exosomes from MCF-7 cells compared to MDA-MB-231 and non-cancerous cells, showing a dependence of the levels of this exo-somal lipid on the ER status of the cell of origin [195]. Another study compared the lipid profile of EVs isolated from TNBC cell lines, D3H2LN and D3H1, with high and low metastatic potential, respectively [196]. The exosomal levels of unsaturated diacylglycerols isolated from high metastatic cells were higher and stimulated angiogenesis through protein kinase D signaling pathway.” (lines 482-495).
- “EVs proteins from SKBR3 (Her2 subtype) and MCF-7 (Luminal A subtype) cells are involved in lipogenesis, glycolysis/gluconeogenesis, and tricarboxylic acid corroborating the important role of altered metabolism in cancer. Moreover, glycolytic enzymes can protect cancer cells from stress by inhibiting apoptosis [200]. Taken together, these data show that EVs cargo is associated with the progression, proliferation, metabolic control, and malignancy of breast cancer (Figure 3).” (lines 523-528)
- “Moreover, it was found that the exosomal lipid composition also contributes to facilitating adherence/internalization in recipient cells [187]“ (lines 480-482)”.
Point 4.3:
- Abstract: line 5; page 3, figure legend 1, line 8- please remove the extra spaces.
Response 4.3:
Thank you for this comment. We apologize for the unnecessary spaces in the organization of the manuscript.
Point 4.4:
- The authors should also discuss about the target specificity of EVs in therapeutic settings.
Response 4.4:
Thank you for pointing this out. In this version, we added a New Section entitled “EVs in oncology precision medicine and treatment: promises and opening questions” to address the therapeutic settings of EVs.
Point 4.5:
- Please add the following references to acknowledge the role of EVs in breast cancer:
a). Ref. 1: Sci Rep. 2018 May 9;8(1):7357. doi: 10.1038/s41598-018-25725-w; PMID: 29743547
b). Ref. 2: Mol Carcinog. 2018 Dec;57(12):1707-1722. doi: 10.1002/mc.22891. Epub 2018 Sep 5; PMID: 30129687
Response 4.5:
We revised these references and we included them in our review. Find these suggestions as refs 15 and 16. We mention them in lines 56-65 and 435-439. Thank you.
Reviewer 5 Report
Deciphering the functional status of breast cancers through the analysis of their extracellular vesicles. By Alexis German Murillo Carrusco et al.
This is a comprehensively written review on extracellular vesicles from breast cancer cells and how they can be exploited for deciphering the functionality of the tumors. Very informative and timely review for breast cancer researchers. I just had one small problem. It appears that part of Fig. 3 was adopted from Biorender but was not properly cited. If this is the case it should be cited as per the requirements of Biorender.
Author Response
Reviewer #5
Point 5.1:
Deciphering the functional status of breast cancers through the analysis of their extracellular vesicles. By Alexis German Murillo Carrusco et al.
This is a comprehensively written review on extracellular vesicles from breast cancer cells and how they can be exploited for deciphering the functionality of the tumors. Very informative and timely review for breast cancer researchers. I just had one small problem. It appears that part of Fig. 3 was adopted from Biorender but was not properly cited. If this is the case it should be cited as per the requirements of Biorender.
Response 5.1:
Thank you for your kind comment and summary of our study. We added the proper citation to Biorender in Figures 1 and 3.
Round 2
Reviewer 2 Report
Well done. No more questions from my side.
-